# Anatomical and biomechanical traits of broiler chickens across ontogeny. Part II. Body segment inertial properties and muscle architecture of the pelvic limb

Heather Paxton[1], Peter G. Tickle[2], Jeffery W. Rankin[1], Jonathan R. Codd[2] and John R. Hutchinson[1]

[1] Structure & Motion Laboratory, Department of Comparative Biomedical Sciences, The Royal Veterinary College, University of London, Hatfield, Hertfordshire, UK
[2] Faculty of Life Sciences, University of Manchester, Manchester, UK

## ABSTRACT

In broiler chickens, genetic success for desired production traits is often shadowed by welfare concerns related to musculoskeletal health. Whilst these concerns are clear, a viable solution is still elusive. Part of the solution lies in knowing how anatomical changes in afflicted body systems that occur across ontogeny influence standing and moving. Here, to demonstrate these changes we quantify the segment inertial properties of the whole body, trunk (legs removed) and the right pelvic limb segments of five broilers at three different age groups across development. We also consider how muscle architecture (mass, fascicle length and other properties related to mechanics) changes for selected muscles of the pelvic limb. All broilers used had no observed lameness, but we document the limb pathologies identified post mortem, since these two factors do not always correlate, as shown here. The most common leg disorders, including bacterial chondronecrosis with osteomyelitis and rotational and angular deformities of the lower limb, were observed in chickens at all developmental stages. Whole limb morphology is not uniform relative to body size, with broilers obtaining large thighs and feet between four and six weeks of age. This implies that the energetic cost of swinging the limbs is markedly increased across this growth period, perhaps contributing to reduced activity levels. Hindlimb bone length does not change during this period, which may be advantageous for increased stability despite the increased energetic costs. Increased pectoral muscle growth appears to move the centre of mass cranio-dorsally in the last two weeks of growth. This has direct consequences for locomotion (potentially greater limb muscle stresses during standing and moving). Our study is the first to measure these changes in the musculoskeletal system across growth in chickens, and reveals how artificially selected changes of the morphology of the pectoral apparatus may cause deficits in locomotion.

Corresponding author
Heather Paxton, hpaxton@rvc.ac.uk

## INTRODUCTION

The poultry industry is a rapidly expanding enterprise (over 870 million broilers slaughtered in the UK; DEFRA 2013 statistics), in which production continues to increase globally by over 130% in some countries (*Scanes, 2007*). In addition to increased production, the broiler chicken has gained an unusual repertoire of anatomical traits, which are repeatedly emphasized in scientific studies and highlight what has developed into a successful breeding programme for obtaining desired production characteristics (e.g., *Havenstein, Ferket & Qureshi, 2003*; *Paxton et al., 2010*; *Paxton et al., 2013*). However, a crossroads has been reached where efficient broiler production is haunted by welfare concerns (*Julian, 1998*; *Mench, 2004*; *Knowles et al., 2008*). Broilers may suffer from heart failure and sudden death syndrome (*Julian, 1998*; *Maxwell & Robertson, 1998*; *Olkowski, 2007*), reduced adaptive immune function (*Cheema, Qureshi & Havenstein, 2003*), leg weakness (see review; *Bradshaw, Kirkden & Broom, 2002*), poor reproductive performance (*Siegel & Dunnington, 1987*; *Hocking, 1993*) and appear to be susceptible to suboptimal management of nutrition and their environment (e.g., *Vestergaard & Sanotra, 1999*; *Kestin et al., 2001*; *Scott, 2002*; *Brickett et al., 2007*; *Buijs et al., 2009*). Unfortunately, whilst the welfare and economic concerns associated with these issues are clear, there is no optimal evidence-based solution that resolves all concerns surrounding broiler chicken production.

To move toward such a solution, a deeper understanding of how broiler body shape and musculoskeletal function develops during growth and how these changes may influence locomotion is required. Part 1 of this series (*Tickle et al., 2014*) characterized how broiler organ and pectoral muscle growth varies with increasing body mass, with a focus on respiratory system development and changes in organ size, highlighting important repercussions to breathing/cardiorespiratory performance. Here, we delve deeper into how these and other anatomical changes have affected the overall size and shape of the broiler and investigate how relevant traits have likely influenced their locomotor abilities.

To help achieve our aim, we detail the muscle architectural properties of the major pelvic limb muscles (identified previously by *Paxton et al., 2010* as the hip, knee and ankle extensors in broilers) and document how these scale with body size. Skeletal muscle is one of the organs that is most adaptable to environmental change (*Lieber, 1986*), and an integral component of locomotion (supporting and powering the movement). Muscle mechanical performance is mostly dependent on a few key architectural properties; namely mass, fascicle length and pennation angle. These parameters are typically used to calculate physiological cross-sectional area (PCSA) and thereby estimate muscular capacity for force-generation (PCSA) versus length change (fascicle length) (*Powell et al., 1984*; *Burkholder et al., 1994*; *Lieber & Friden, 2000*). For broilers, muscle architecture data on the broiler currently exist for animals at six weeks of age (see *Paxton et al., 2010*), but how these properties change across growth in the broiler is unknown. This study therefore focuses on the scaling (i.e., size related, isometric or allometric; *Biewener, 1989*; *Alexander et al., 1981*; *Alexander & Ker, 1990*) relationships of select pelvic limb muscles to reveal changes in individual muscle characteristics of individual muscles from hatching to slaughter age (~six weeks). We also consider how the pelvic limb bones (femur, tibiotarsus

and tarsometatarsus) scale with body size in the broiler chicken, as effective scaling of the hindlimb bones can reduce the rate at which stress increases with body mass (*McMahon, 1973*; *McMahon, 1975*; *Garcia & da Silva, 2004*).

In addition, differences in limb orientation and motion, and overall gait dynamics, among avian taxa can be partially attributed to variation in body centre of mass (CoM) position (*Gatesy & Biewener, 1991*; *Abourachid, 1993*; *Hutchinson, 2004*). *Manion (1984)* estimated CoM position for chickens across ontogeny (5–19 days) and noted a cranioventral shift and a subsequent change in limb orientation during standing and walking (more flexed during standing, but more extended during walking). The broiler CoM position has been addressed in preliminary computational analyses by *Allen, Paxton & Hutchinson (2009)* and was found to shift caudodorsally. Other previous research suggests that broilers have a more cranially positioned CoM induced by a large pectoral muscle mass (*Abourachid, 1993*; *Corr et al., 2003a*). To better understand the influence of CoM on locomotor ability in broilers, we therefore quantify the variation in CoM position (3D) across ontogeny and relate this variation to anatomical changes documented here and in Part I of this study (*Tickle et al., 2014*).

All broilers used here are considered 'normal'—i.e., no observed lameness, but we document the limb pathologies (identified post mortem) within our study populations. This is essential because the pathological changes in affected birds do not often relate to walking ability (assessed using gait score and force plate measurements) (*Sandilands et al., 2011*). We quantify the inertial properties (mass, centre of mass and radius of gyration) of each limb and major body segment (Fig. 1), because these help reveal basic locomotor habits (e.g. limb tapering—i.e., a proximal to distal reduction in muscle mass reflects a specialization for power versus force development; *Alexander et al., 1981*; *Pasi & Carrier, 2003*; *Hutchinson, 2004*; *Payne et al., 2005*; *Smith et al., 2006*). Both centre of mass position and radius of gyration are also good descriptors of body area distribution and resistance to rotational movements (*Kilbourne, 2013*). Importantly, inertial properties in chickens and more specifically broilers are almost completely unstudied. Only a few studies have documented the inertial properties of other ground running birds (e.g., emus, *Goetz et al., 2008*; guinea fowl, *Daley, Felix & Biewener, 2007*; *Rubenson & Marsh, 2009*; quail, *Andrada et al., 2013*; lapwing, *Nyakatura et al., 2012*). This study not only provides a novel insight into characteristics that influence broiler locomotor function, but also provides the inertial properties necessary to develop models for dynamic analyses of movement, which have been highly successful in improving our understanding of human pathological gait (e.g., *Steel, Van der Krogt & Delp, 2012*; *Fey, Klute & Neptune, 2013*; *Allen, Kautz & Neptune, 2013*).

## MATERIALS AND METHODS

### Birds

Fresh male cadavers of a commercial broiler strain at different ages (days 1, 13, 29, 32 and 40; Table 1) were used for the hind limb muscle analysis and then a further five broilers of the same commercial strain at approximately 14, 28 and 42 days of age (2, 4

**Peer**J

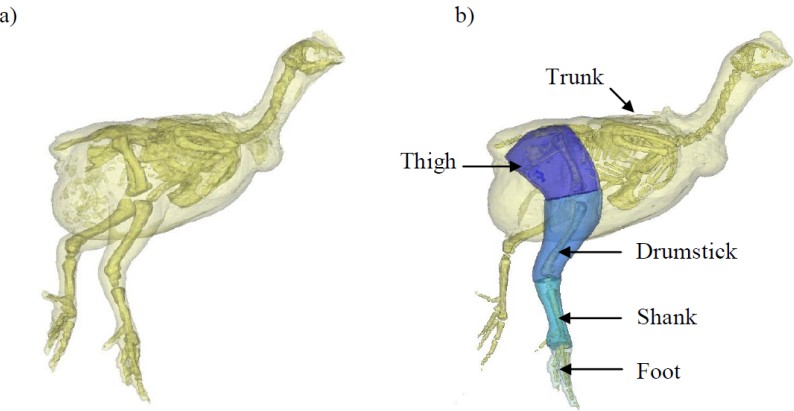

**Figure 1 A 3D model representation of the broiler showing the body and pelvic limb segments.** A 3D model representation of the broiler's body and its corresponding segments created within MIMICS software. Each chicken was placed in the same upright position with their left side resting on a radiolucent cushion during scanning. (A) shows the whole body and skeleton produced within this software and (B) shows the trunk (leg flesh removed) and pelvic limb segments. The translucent outline represents the 'virtual flesh' of the models used to estimate the inertial properties (mass, centre of mass and radius of gyration) of each segment.

**Table 1 Subject data.** Data represented here are for the broiler chickens used only for muscle architecture and are means ± standard deviation. These data form part of the mean data presented in Table 2, Part I of this study.

| Age (days) | Sample size (*n*) | Body mass (kg) |
|---|---|---|
| 1 | 10 | 0.044 ± 0.01 |
| 13 | 10 | 0.431 ± 0.08 |
| 29 | 5 | 1.542 ± 0.05 |
| 32 | 5 | 1.738 ± 0.08 |
| 40 | 7 | 2.452 ± 0.09 |

and 6 weeks old) were used to calculate the centre of mass (CoM) of the whole body, trunk and the pelvic limb segments—i.e., the thigh, drumstick, shank and foot (Fig. 1). Hatchlings (day 1) are not included in these CoM data, since our focus is in later stages of their development when breast muscle growth is more evident and leg health issues are apparent. These chickens had been previously killed by cervical dislocation. Pelvic limb bone dimensions (femur, tibiotarsus and tarsometatarsus) were also recorded at 2, 4 and 6 weeks of age. These data are from the same chickens used in Part I (*Tickle et al., 2014*), which focuses on anatomy of the musculoskeletal respiratory apparatus and changes in body and organ size across ontogeny. As our data are cross-sectional rather than longitudinal, this study approximates an ontogenetic analysis, via inference from comparisons among individuals. All chickens were raised in a commercial setting under similar management guidelines and were not outwardly influenced by a laboratory

**Table 2 Incidence of pathology across growth in broiler chickens.** Data are presented as percentage of total study population. Bacterial chondronecrosis (BCO) and tibial dyschondroplasia (TD) were marked as present or absent. The severity of these abnormalities was not considered. Where present, valgus/varus deformities (VVD) were classified as mild (10–25°), moderate (25–45°) or severe (>45°) following the methods described by *Leterrier & Nys (1992)*. Tibial rotation (RT) above 20° was considered abnormal. Pathologies may have been recorded in one pelvic limb or both pelvic limbs of the individual birds; this is not distinguished here.

| Age (days) | BCO (femur) | BCO (tibiotarsus) | TD (present) | VVD | | | | RT (abnormal) |
|---|---|---|---|---|---|---|---|---|
| | | | | Normal | Mild | Moderate | Severe | |
| 14 | 63 | 53 | 43 | 100 | 0 | 0 | 0 | 33 |
| 28 | 75 | 93 | 57 | 100 | 0 | 0 | 0 | 15 |
| 42 | 88 | 97 | 24 | 55 | 42 | 3 | 0 | 16 |

setting. Full ethical approval for this experiment was granted by the RVC Ethics Committee (approval URN No. 2008-0001) under a Home Office license.

## Pathology

All birds were macroscopically evaluated to establish the incidence of leg pathologies (marked as present/absent unless otherwise stated; Table 2). Each chicken was examined for bacterial chondronecrosis with osteomyelitis (BCO) in the femur and tibiotarsus, tibial dyschondroplasia (TD), tibial rotation (RT) and valgus/varus deformities (VVD). Tibial rotation above 20° was considered abnormal. Similar to other studies (e.g., *Shim et al., 2012*) VVD was classified as mild, moderate or severe following the methods described by *Leterrier & Nys (1992)*.

## Muscle architecture

The left pelvic limb from each individual (refer to Table 1) was dissected, with fourteen specific muscle–tendon units (see Table 3 for muscle names and abbreviations) being identified and systematically removed. Muscles identified included the main hip, knee and ankle extensors, which are involved in limb support during the stance phase and have been shown to have either massive, short-fibred muscles favouring greater force-generating capacity or have long parallel-fibred muscles for fast contraction (*Paxton et al., 2010*), as in many other terrestrial species. Architectural measurements taken included muscle mass ($M_m$; tendon removed), which was measured on an electronic balance ($\pm 0.001$ g), muscle fascicle length ($L_f$; $\pm 1$ mm), and fibre pennation angle ($\theta$), where appropriate ($\pm 1°$). In order to account for variation of fascicle length and pennation angle within a muscle, $L_f$ and $\theta$ were calculated as the mean of five measurements made across each muscle. Physiological cross-sectional area (PCSA) was calculated for each muscle (Eq. (1); *Sacks & Roy, 1982*) from these measurements according to the equation:

$$PCSA = M_m \cos\theta (\rho L_f)^{-1}. \tag{1}$$

Muscle tissue density ($\rho$) was assumed to be 1.06 g cm$^{-3}$, the standard value for mammalian and avian muscle (*Mendez & Keys, 1960*; *Paxton et al., 2010*).

**Table 3** The major muscles of the chicken pelvic limb.

| Muscle | Abbreviation |
| --- | --- |
| M. iliotibialis cranialis | IC |
| M. iliotibialis lateralis | IL |
| M. gastrocnemius pars lateralis | GL |
| M. gastrocnemius pars medialis | GM |
| M. fibularis longus | FL |
| M. iliotrochantericus caudalis | ITC |
| M. femorotibialis medialis | FMT |
| M. iliofibularis | ILFB |
| M. flexor cruris lateralis pars pelvica | FCLP |
| M. flexor cruris medialis | FCM |
| M. caudofemoralis pars caudalis | CFC |
| M. tibialis cranialis caput femorale | TCF |
| M. puboischiofemoralis pars medialis | PIFM |
| M. puboischiofemoralis pars lateralis | PIFL |

## Centre of mass and inertial properties

Computed tomography (CT) scans were acquired of five male bird cadavers from each group. The cadavers were scanned with a GE Lightspeed 8-detector scanner at 100 mA and 120 kVp X-ray beam intensity using a 1 mm CT slice thickness. In order to minimise postural effects on CoM estimates, the birds were placed in the same upright position with their left side resting on a radiolucent cushion for scanning and limbs posed as similarly as possible (Fig. 1A). Mimics 14.12 imaging software (Materialise; Leuven, Belgium) was then used to segment the resulting DICOM image files, creating a 3D representation of the skeleton, body and the pelvic limb segments (trunk, thigh, drumstick, shank and the foot; Fig. 1B). We used predefined thresholds set for bone and flesh, with some manual adjustment from those baseline values as appropriate to ensure smooth, plausible rendering. Custom software (*Hutchinson, Ng-Thow-Hing & Anderson, 2007*; *Allen, Paxton & Hutchinson, 2009*) was then used to estimate the whole body/segment CoM. This method allows accurate specification of the CoM relative to any user defined point on the body or segment. These points are easily identified using bony landmarks visible on the CT images and are essential for putting CoM position in the context of gross morphology, which is necessary for biomechanical analyses. Using the 3D model representations of the skeleton, whole body CoM position was quantified relative to the right hip, trunk CoM was taken relative to the pelvis (centre point between the hips on the pelvis), and for the remaining segments, CoM position was expressed relative to the proximal end of the bone (Fig. 2). Three-dimensional coordinates ($x$ [craniocaudal], $y$ [dorsoventral] and $z$ [mediolateral]) for the CoM were then recorded. Whole body CoM position was expressed as a percentage of femur length (see *Allen, Paxton & Hutchinson, 2009*).

Segment anatomical properties that we measured and present here include segment mass ($m$; as % body mass), segment length ($L$; see Fig. 2), centre of mass (as % segment

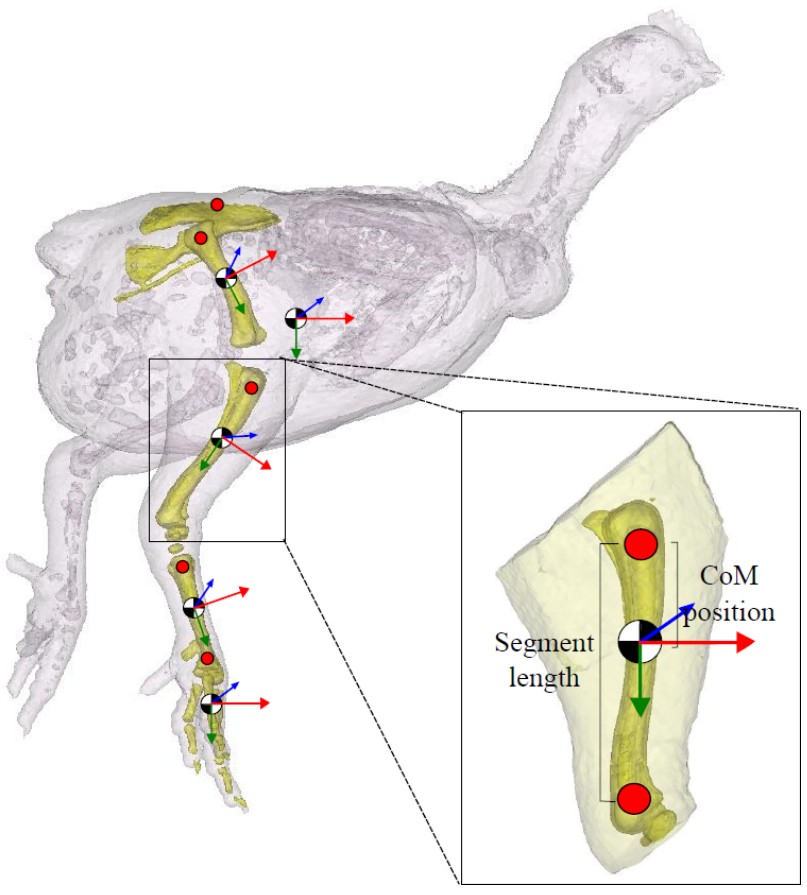

**Figure 2 Segment inertial properties.** The pelvis, femur, tibiotarsus, tarsometatarsus and the bones of the foot are highlighted in this 3D model. Centre of mass (CoM) position is shown (black/white circle; approximate position given). Trunk CoM was taken relative to the pelvis (centre point between the hips on the pelvis), and for the remaining segments, CoM positionwas expressed relative to the proximal end of the bone (red markers shown). The local anatomical coordinate system for each segment is given ($x$ (red), $y$ (green) and $z$ (blue)). Segment length (excluding the pelvis) is defined as the distance between the proximal and distal marker on the segment, as shown.

length), and radius of gyration ($r$; as % segment length), which are essential information required to calculate the moments of inertia (kg m$^2$; Eq. (3)) and subsequently complete the set of inertial properties required for biomechanical analyses:

$$R = (Im^{-1})^{0.5}. \tag{2}$$

The radius of gyration (% segment length; Eq. (2)) was calculated using the principal moment of inertia ($I$) and mass of the segment, both estimated using the custom software. The mean difference between the dissected segment mass recorded and that estimated were small ($<$5%). These data also provide further information on muscle mass distribution within the limb.

$$I = m(rL)^2. \tag{3}$$

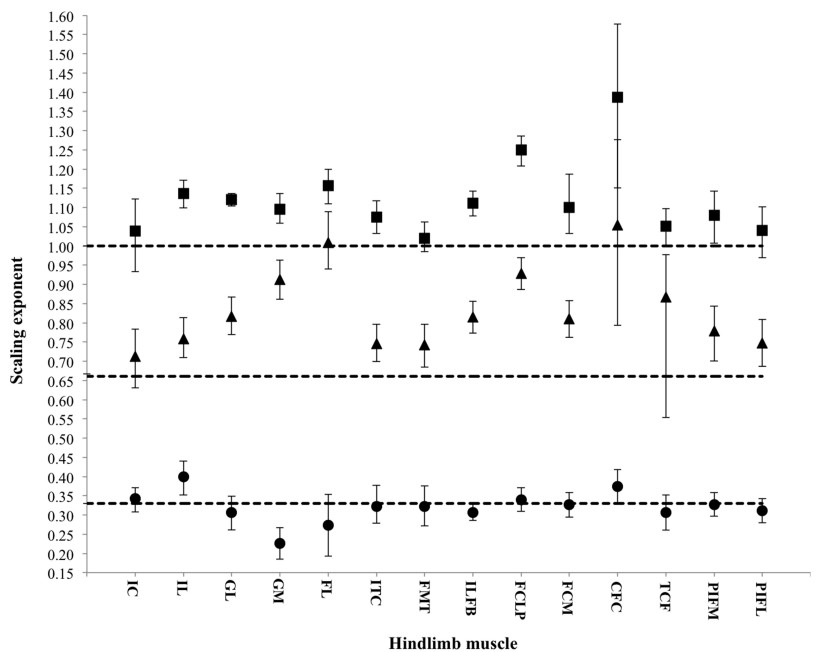

**Figure 3 Scaling exponents of pelvic limb muscle properties as a function of increasing body mass.** Symbols indicate the regression slope for muscle mass (squares), physiological cross-sectional area (PCSA; triangles) and muscle fascicle length (circles). Error bars represent upper and lower 95% confidence intervals. Dashed lines represent, expected values for isometric (directly proportional) scaling of muscle properties with body mass ($y = 1.0$: muscle mass, $y = 0.67$: PCSA, $y = 0.33$: muscle fascicle length).

## Bone scaling dataset

Data from Part I of this study were also used for this analysis. The individual bone lengths of the left pelvic limb were recorded ($\pm 1$ mm) for each of four age groups: day 1 ($n = 10$), ~14 days ($n = 20$), ~28 days ($n = 19$), ~42 days ($n = 19$). Total leg length was defined as the sum of the individual pelvic limb bone lengths. Limb bone proportions were calculated as a percentage of total leg length.

## Statistical analysis

All of our analyses take into account differences in body size across age groups. To analyse the muscle architecture data, the linear relationship between log-transformed body mass and muscle mass, fascicle length and PCSA were examined using the reduced major-axis regression (RMA) function in the statistical program PAST (*Hammer, Harper & Ryan, 2001*). RMA regression was appropriate for analysis of these morphological characters because error in both *x* and *y* variables is considered (*Rayner, 1985*; *Sokal & Rohlf, 1995*). Upper and lower 95% confidence intervals (CIs) and the $R^2$ value for each regression line were calculated to quantify the variation around the mean (Fig. 3, Table 4). Isometric scaling was assumed when the scaling exponent $\pm$95% CIs overlapped the expected value. Reduced major-axis regression was also used to examine the linear relationship between log-transformed body mass and bone length. This was done using custom Matlab (The Mathworks, Nattick, MA, USA) software code. The 95% CIs and the $R^2$ value for each

**Table 4 RMA regression analyses.** Data here are the RMA regression analyses that were used to determine the linear relationships between pelvic limb muscle architecture and body mass. Expected regression slopes for isometric growth are given for each parameter. Symbols next to each calculated regression slope indicate isometric growth (=), positive allometry (+; emphasized in bold) or negative allometry (−; emphasized in italics). All regressions were significant ($p < 0.05$).

| Muscle | $M_m$ expected slope: 1.00 | | | | PCSA expected slope: 0.67 | | | | $L_f$ expected slope: 0.33 | | | |
|---|---|---|---|---|---|---|---|---|---|---|---|---|
| | Slope | Lower 95% CI | Upper 95% CI | $R^2$ | Slope | Lower 95% CI | Upper 95% CI | $R^2$ | Slope | Lower 95% CI | Upper 95% CI | $R^2$ |
| IC | 1.038 (=) | 0.933 | 1.123 | 0.961 | 0.712 (=) | 0.630 | 0.783 | 0.938 | 0.342 (=) | 0.308 | 0.372 | 0.923 |
| IL | **1.136**(+) | 1.099 | 1.171 | 0.994 | **0.759**(+) | 0.709 | 0.814 | 0.968 | **0.399**(+) | 0.352 | 0.440 | 0.938 |
| GL | **1.121**(+) | 1.104 | 1.136 | 0.998 | **0.817**(+) | 0.769 | 0.868 | 0.977 | 0.306 (=) | 0.261 | 0.349 | 0.859 |
| GM | **1.095**(+) | 1.059 | 1.136 | 0.992 | **0.912**(+) | 0.861 | 0.963 | 0.977 | *0.226*(−) | 0.185 | 0.268 | 0.801 |
| FL | **1.156**(+) | 1.109 | 1.199 | 0.988 | **1.009**(+) | 0.940 | 1.089 | 0.945 | 0.274 (=) | 0.193 | 0.354 | 0.420 |
| ITC | **1.075**(+) | 1.032 | 1.118 | 0.989 | **0.746**(+) | 0.699 | 0.796 | 0.969 | 0.322 (=) | 0.279 | 0.377 | 0.844 |
| FMTM | 1.021 (=) | 0.985 | 1.062 | 0.984 | **0.743** (+) | 0.685 | 0.796 | 0.948 | 0.322 (=) | 0.271 | 0.375 | 0.758 |
| ILFB | **1.112**(+) | 1.078 | 1.143 | 0.994 | **0.815**(+) | 0.773 | 0.856 | 0.981 | 0.307 (=) | 0.286 | 0.330 | 0.961 |
| FCLP | **1.250**(+) | 1.208 | 1.286 | 0.992 | **0.929**(+) | 0.887 | 0.969 | 0.979 | 0.339 (=) | 0.309 | 0.371 | 0.924 |
| FCM | **1.101**(+) | 1.032 | 1.187 | 0.958 | **0.811**(+) | 0.762 | 0.857 | 0.974 | 0.326 (=) | 0.294 | 0.358 | 0.938 |
| CFC | **1.387**(+) | 1.151 | 1.578 | 0.926 | **1.054**(+) | 0.793 | 1.277 | 0.869 | 0.374 (=) | 0.329 | 0.419 | 0.881 |
| TCF | 1.052 (=) | 1.000 | 1.097 | 0.979 | 0.867 (=) | 0.553 | 0.977 | 0.887 | 0.306 (=) | 0.260 | 0.352 | 0.720 |
| PIFM | 1.079 (=) | 1.007 | 1.143 | 0.969 | **0.779**(+) | 0.701 | 0.843 | 0.933 | 0.327 (=) | 0.297 | 0.358 | 0.893 |
| PIFL | 1.040 (=) | 0.969 | 1.102 | 0.975 | **0.748**(+) | 0.687 | 0.808 | 0.952 | 0.311 (=) | 0.280 | 0.343 | 0.918 |

regression line were also calculated (Fig. 4). Similar to the statistical analysis used for the muscle architecture data, isometric scaling was assumed when the scaling exponent ±95% CIs overlapped the expected value.

For CoM values and segment anatomical properties, the statistics used followed those set out in Part I of this study. In brief, a Shapiro–Wilk normality test, in combination with considering subsequent PP and QQ plots of the data, and Levene's test for equal variances were used to test the assumptions of a one-way analysis of variance (ANOVA). An ANOVA test was then conducted with a subsequent Bonferroni post-hoc test to check for differences among the three age groups. If equal variances were violated, the Welch statistics are reported in conjunction with the results of a subsequent Games-Howell post-hoc test (Table 8).

## RESULTS

### Pathology

There were a number of pathological changes in the study population at all stages of development (see Table 2). Bacterial chondronecrosis with osteomyelitis (BCO) was present in all populations both in the femur and proximal tibia. There was an increased incidence of BCO in the femur as the broilers aged, with 88% of the six week old chickens affected. Tibial dyschondroplasia was present in all populations with no apparent correlation with age (average 41%). Rotated tibia was more prevalent in the younger birds (~33% of the study population). Valgus angulation of the lower limb was seen only in

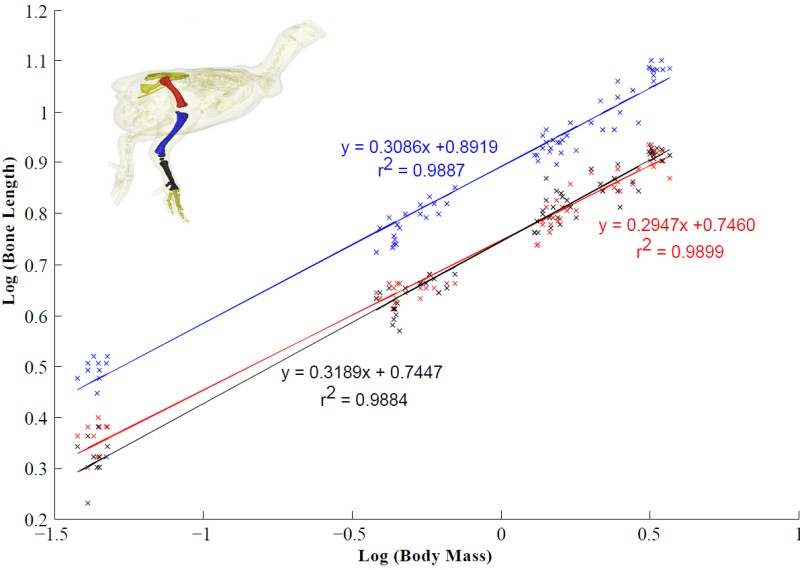

**Figure 4 Scaling relationship of the pelvic limb bones (femur, tibiotarsus and tarsometatarsus).** 95% confidence intervals (CIs) for the femur, tibiotarsus and tarsometatarsus are $0.2947 \pm 0.0104$, $0.3086 \pm 0.0114$ and $0.3189 \pm 0.0120$ respectively (scaling exponent $\pm$ CI). Isometric scaling of the tibiotarsus and tarsometatarsus is concluded because the 95% CIs overlapped the expected value (0.33). The femur scaled with slight negative allometry.

the six-week old chickens, with approximately 45% of these chickens affected. No varus deformities were observed.

## Muscle architecture

Across ontogeny, the masses of the major hip, knee and ankle extensor muscles generally scale with positive allometry, increasing in relative size as broilers grew (Table 4). The exceptions were the FMTM (M. femorotibialis medialis), TCF (M. tibialis cranialis caput femorale), PIF (M. puboischiofemoralis) and IC (M. iliotibialis cranialis), which did not change significantly with body mass. Interestingly, fascicle length, in general, did not change with body mass, but scaled isometrically. However, the IL (M. iliotibialis lateralis) muscle showed an increase in fascicle length, whereas the GM (M. gastrocnemius medialis) showed a decrease in fascicle length. Despite these changes in fascicle length, PCSA appeared to increase allometrically across ontogeny in the majority of the pelvic limb muscles. The IC and TCF, however, scaled more in accordance with isometry.

## Whole body centre of mass

Whole body CoM moved caudodorsally until 28 days of age. By 42 days of age there was a significant cranial shift in CoM position, moving ~10% cranially and ~30% more dorsally in the last two weeks approaching slaughter age (Table 5).

## Segment properties

There was a 50-fold range in body mass of our sample of broiler chickens (Table 1), ranging from ~0.04 kg at one day old to ~2.4 kg at 40 days of age. Trunk mass contribution to

**Table 5  Whole body centre of mass position.** Data represented are means ± standard deviation. Centre of mass (CoM) position is calculated as a percentage of femur length and is expressed relative to the right hip joint of the chicken. Craniocaudal and dorsoventral positions are shown. Data with no common superscript differ significantly at the 0.05 level.

| Age group | CoM Position (% femur length) | |
|---|---|---|
| | Craniocaudal | Dorsoventral |
| 14 days | $90.6 \pm 10.7$[1] | **$89.8 \pm 13$[1]** |
| 28 days | $68.3 \pm 5.0$[1,2] | $55.4 \pm 17.7$[2] |
| 42 days | **$76.6 \pm 12.2$[2]** | $28.2 \pm 19.5$[2] |

**Table 6  Pelvic limb segment inertial properties.** Data represented are means ± standard deviation. Centre of mass position (CoM) is located relative to the proximal end of the segment (trunk CoM is relative to the centre line between the hips), and is shown along the craniocaudal (for trunk) or proximodistal (for limbs) axis (Fig. 2). Data in a column with no common superscript differ significantly at the 0.05 level.

| Segment | Age group | Segment mass | CoM position | Radius of Gyration (% segment length) | | |
|---|---|---|---|---|---|---|
| | | (% body mass) | (% segment length) | $x$ | $y$ | $z$ |
| Trunk | 14 days | $74.6 \pm 1.7^a$ | $19.5 \pm 2.5$ | $23.2 \pm 1.0$ | $41.1 \pm 0.3^b$ | $35.0 \pm 1.9^a$ |
| | 28days | $78.0 \pm 1.8^{a,b}$ | $20.1 \pm 4.3$ | $24.0 \pm 1.2$ | $40.0 \pm 1.7^b$ | $37.1 \pm 1.8^{a,b}$ |
| | 42 days | $81.7 \pm .2^b$ | $15.3 \pm 2.7$ | $33.3 \pm 12.0$ | **$37.1 \pm 1.8^a$** | $37.8 \pm 0.8^b$ |
| Thigh | 14 days | $5.19 \pm 1.4^a$ | **$43.7 \pm 4.3^a$** | $47.2 \pm 8.5$ | $44.1 \pm 7.4$ | **$49.6 \pm 14.5$** |
| | 28days | $5.22 \pm 0.4^a$ | **$38.1 \pm 2.0^b$** | $48.3 \pm 6.2$ | $40.6 \pm 4.2$ | **$52.1 \pm 8.8$** |
| | 42 days | **$8.21 \pm 1.0^b$** | **$28.7 \pm 2.4^c$** | $46.2 \pm 7.0$ | $42.5 \pm 4.3$ | $52.3 \pm 7.1$ |
| Drumstick | 14 days | **$3.9 \pm 0.3^a$** | $30.1 \pm 14.2$ | **$49.1 \pm 6.3$** | **$24.9 \pm 2.9$** | **$51.0 \pm 6.1^a$** |
| | 28days | **$4.6 \pm 0.5^b$** | **$32.7 \pm 2.3$** | **$40.0 \pm 5.3$** | **$25.6 \pm 4.0$** | **$40.2 \pm 7.1^{a,b}$** |
| | 42 days | **$5.66 \pm 0.2^c$** | **$24.2 \pm 4.2$** | $34.0 \pm 12.4$ | **$21.2 \pm 11.7$** | **$34.3 \pm 12.2^b$** |
| Shank | 14 days | **$0.98 \pm 0.1^a$** | **$36.1 \pm 10.3^a$** | $50.5 \pm 1.2^a$ | $18.1 \pm 0.2$ | $50.8 \pm 1.1^a$ |
| | 28days | $1.02 \pm 0.1^{a,b}$ | $17.3 \pm 5.5^b$ | $49.8 \pm 2.3^a$ | $17.2 \pm 1.2$ | $50.2 \pm 2.5^a$ |
| | 42 days | $1.23 \pm 0.2^b$ | $22.6 \pm 6.3^b$ | **$38.0 \pm 4.1^b$** | $23.3 \pm 20.7$ | **$37.1 \pm 2.7^b$** |
| Foot | 14 days | $0.63 \pm 0.03^a$ | $38.6 \pm 4.3^a$ | $34.3 \pm 0.9$ | $35.2 \pm 1.8^b$ | $23.3 \pm 1.4$ |
| | 28days | $0.61 \pm 0.07^a$ | $28.4 \pm 7.3^a$ | $31.0 \pm 2.3$ | **$21.2 \pm 6.3^a$** | $28.0 \pm 4.5$ |
| | 42 days | **$0.91 \pm 0.12^b$** | **$51.9 \pm 9.1^b$** | $38.7 \pm 16.4$ | $29.9 \pm 3.9^b$ | $29.7 \pm 21.4$ |

whole body mass increased across ontogeny and was ∼5% larger at 42 days of age relative to the youngest group. Similarly, a significantly larger relative thigh muscle mass was found in older birds (by ∼3% body mass) compared to the younger broilers. Drumstick mass increased significantly with age whereas shank mass remained unchanged. Between 28 and 42 days, the chickens' foot mass also increased significantly by ∼30% (see Table 6).

Trunk CoM moved caudally between 28 and 42 days, and the thigh, drumstick and shank CoM moved to more proximal positions (Table 5). In contrast, the foot's CoM moved more distally at 42 days of age.

The radii of gyration about the axes of long-axis rotation (Table 6) experienced an increase in the foot segments of the six week old broilers and remained unchanged in the thigh, drumstick and shank. However, the radius of gyration of the whole trunk segment

**Table 7 Pelvic limb bone segment dimensions.** Data presented here are for the left pelvic limb only, and are means ± standard deviation. Total leg length is the sum of the individual pelvic limb bones. Absolute values for leg length are presented here, but normalized values (divided by body mass$^{1/3}$) were used for the statistical analysis to compare how leg length changed across ontogeny (see Fig. 4 for scaling relationship). Data in a column with no common superscript differ significantly at the 0.05 level.

| Age group | Sample size | Leg length (cm) | Individual bones (% leg length) | | |
|---|---|---|---|---|---|
| | | | Femur | Tibiotarsus | Tarsometatarsus |
| 1 day | 10 | **$7.5 \pm 0.44^a$** | $30.4 \pm 0.68$ | $41.5 \pm 1.8$ | $28.0 \pm 1.4$ |
| 14 days | 19 | $14.9 \pm 0.93^b$ | $29.9 \pm 0.6$ | $40.9 \pm 1.1$ | $29.2 \pm 1.0$ |
| 28 days | 19 | $21.4 \pm 1.3^b$ | $29.3 \pm 1.1$ | $40.4 \pm 1.5$ | $30.2 \pm 1.1$ |
| 42 days | 20 | $26.9 \pm 2.2^b$ | $29.0 \pm 1.1$ | $42.0 \pm 1.4$ | $29.0 \pm 0.7$ |

showed a decrease for long-axis rotation—i.e., a lower resistance to yaw. The radii of gyration in the parasagittal plane, decreased in the drumstick and shank across broiler ontogeny, but progressively increased in the foot. There was also a relative reduction of $r$% about the axis of abduction/adduction rotation. Radii of gyration for the thigh remained unchanged.

### Bone scaling

Tibiotarsus and tarsometatarsus length scaled isometrically with body mass, whereas femur length scaled with slight negative allometry (Fig. 4). As a result, there was a relative increase in total leg length from 14 days to 28 days. Limb length remained unchanged between four and six weeks (Table 7). The femur accounts for ∼30% of total leg length and the tibiotarsus accounts for ∼41% of leg length. The tarsometatarsus is relatively shorter than the other pelvic limb bones accounting for ∼29% of total leg length.

## DISCUSSION

The genetic success of the modern broiler and the subsequent changes to the morphology of broiler chickens have been well documented, in order to determine the lines' commercial performance (e.g., *Gous et al., 1999*) and compare both growth responses and physiological adaptations resulting from distinctive selection pressures (e.g., *Havenstein, Ferket & Qureshi, 2003*; *Reddish & Lilburn, 2004*; *Schmidt et al., 2009*). A marked change in total pectoral muscle mass of the commercial broiler is a common finding of all these prior studies. Similarly, we found this mass to represent ∼20% of total body mass in slaughter age chickens (see Part I; *Tickle et al., 2014*). Part I revealed how enlarged pectoral muscle mass, among other anatomical changes, may compromise the efficacy of the respiratory apparatus. Here we show how these changes influence the locomotor ability of the broiler.

### Pathology

Leg weakness in broilers comprises not only nonspecific gait problems and lower activity levels, but also a wide range of disorders that are generally classified as infectious, degenerative, or developmental (for a review see *Bradshaw, Kirkden & Broom, 2002*).

**Table 8 Levene's test and ANOVA results.** Degrees of freedom $= (df_{between}, df_{within})$. Where the assumption of equal variances cannot be met (significant Levene's test result), the Welch statistics are reported.

| | Levene's test | Degrees of freedom | F | P |
|---|---|---|---|---|
| **Bone lengths** | | | | |
| Total Leg Length | <0.001 | 3, 34.7 | 968.5 | <0.001 |
| **Segment properties** | | | | |
| Whole Body CoM (cranial-caudal) | 0.435 | 2, 12 | 6.629 | 0.011 |
| Whole Body CoM (Dorsal-ventral) | 0.475 | 2, 11 | 16.729 | <0.001 |
| **Trunk** | | | | |
| Mass | 0.295 | 2, 12 | 11.638 | 0.002 |
| CoM position | 0.193 | 2, 12 | 3.227 | 0.076 |
| Radius of gyration ($x$) | <0.001 | 2, 7.1 | 2.088 | 0.194 |
| Radius of gyration ($y$) | 0.004 | 2, 4.881 | 9.732 | 0.020 |
| Radius of gyration ($z$) | 0.171 | 2, 12 | 4.551 | 0.034 |
| **Thigh** | | | | |
| Mass | 0.180 | 2, 12 | 13.65 | 0.001 |
| CoM position | 0.547 | 2, 12 | 30.675 | <0.001 |
| Radius of gyration ($x$) | 0.555 | 2, 11 | 0.108 | 0.899 |
| Radius of gyration ($y$) | 0.508 | 2, 12 | 0.514 | 0.611 |
| Radius of gyration ($z$) | 0.103 | 2, 12 | 0.099 | 0.907 |
| **Drumstick** | | | | |
| Mass | 0.002 | 2, 6.885 | 69.702 | <0.001 |
| CoM position | 0.054 | 2, 12 | 1.261 | 0.318 |
| Radius of gyration ($x$) | 0.358 | 2, 12 | 3.902 | 0.050 |
| Radius of gyration ($y$) | 0.270 | 2, 12 | 1.024 | 0.388 |
| Radius of gyration ($z$) | 0.441 | 2, 12 | 4.533 | 0.034 |
| **Shank** | | | | |
| Mass | 0.583 | 2, 12 | 4.820 | 0.029 |
| CoM position | 0.503 | 2, 12 | 7.985 | 0.006 |
| Radius of gyration ($x$) | 0.129 | 2, 12 | 31.746 | <0.001 |
| Radius of gyration ($y$) | 0.012 | 2, 5.542 | 1.297 | 0.695 |
| Radius of gyration ($z$) | 0.301 | 2, 12 | 59.342 | <0.001 |
| **Foot** | | | | |
| Mass | 0.295 | 2, 12 | 18.969 | <0.001 |
| CoM position | 0.502 | 2, 12 | 13.332 | 0.001 |
| Radius of gyration ($x$) | 0.027 | 2, 6.047 | 4.376 | 0.465 |
| Radius of gyration ($y$) | 0.189 | 2, 12 | 1.920 | 0.001 |
| Radius of gyration ($z$) | 0.027 | 2, 5.831 | 2.367 | 0.717 |

The most common disorders include bacterial chondronecrosis with osteomyelitis (BCO), angular and torsional deformities (e.g. valgus-varus (VVD) and rotated tibia (RT)) and tibial dyschondroplasia (TD). With the exception of VVD, these conditions were observed in birds at all developmental stages in this study. BCO was common in the broilers (Table 2), far exceeding previous estimates of infection in commercial flocks

(approximately 0.5% *McNamee et al., 1998*). Incidence of BCO in the femur peaked at 88% of 42-day-old birds. This variation is likely due to differences in how these estimates of infection were determined. Femoral head separation (FHS; epiphyseolysis) dominated in our study's birds, which we concur should be attributed to underlying traumatic (osteochondrosis) or infectious (osteomyelitis) femoral head pathology (following *Wideman et al., 2013*). FHS is often shown separately in studies to show the progression of the disease (e.g., *Wideman et al., 2012*; *Wideman & Pevzner, 2012*; *Wideman et al., 2013*), but here we are purely interested in the presence or absence of the condition. Furthermore, the high percentage of birds we found to have BCO may reflect a predisposition to the condition or differences in husbandry practices (*McNamee & Smyth, 2000*). Increasing occurrence of BCO over development is consistent with previous results that identified peak incidence at around five weeks of age (*McNamee et al., 1999*). Increasing incidence over development may reflect increased stresses acting on the bone, which are thought to contribute to BCO (*Wideman et al., 2013*). Similarly, BCO in the tibiotarsus was widespread and increased with age, showing a peak at 42 days (Table 2). BCO causes lesions in the load-bearing growth plates of the femur and tibiotarsus, so rapid growth and weight gain may be an aggravating factor when bacterial infection is present. Considering the widespread incidence of BCO in birds that were otherwise deemed healthy, bone lesions are a significant problem affecting welfare standards in broiler chickens.

TD commonly leads to growth plate abnormalities, infections and tibial deformation (*Lynch, Thorp & Whitehead, 1992*) but, similar to BCO, TD does not necessarily induce lameness of sufficient severity to impair walking ability (*Pattison, 1992*). TD has been found to occur between 2 and 8 weeks of age (*Lynch, Thorp & Whitehead, 1992*) and our findings are consistent with this observation (Table 2). However, considerable variation exists in the reported prevalence of TD, ranging in 42 day old birds from approximately 2% (*Shim et al., 2012*; *Siller, 1970*) to 50% (*Prasad, Hairr & Dallas, 1972*; *Sauveur & Mongin, 1978*; *Vaiano, Azuolas & Parkinson, 1994*) of total flock population. Incidence of TD in this study was relatively high and occurred in all age groups, peaking at 28 days (57%). However, determining why the incidence of TD is high in this study is difficult because the condition reflects a complicated interaction of contributing factors, including dietary deficiencies, toxins, genetic predisposition and rapid growth rate (*Julian, 1998*; *Orth & Cook, 1994*; *Shim et al., 2012*).

Valgus-varus deformities (VVD) were observed in 42-day-old birds, but was not present in younger broilers (Table 2). Comparable reports indicate that VVD occurs with varying incidence, affecting as few as 0.5% (*Julian, 1984*) to 30–40% of birds in a flock (*Leterrier & Nys, 1992*; *Shim et al., 2012*). The prevalence of VVD in this study fell near the high end of the reported range, with 45% of birds at 42 days of age observed to have mild or moderate VVD. Our observation that symptoms of VVD occur only in older broilers is consistent with the progressive nature of this deformity (*Julian, 1984*; *Julian, 2005*; *Shim et al., 2012*). However, no deformity was seen in 28-day-old birds, which is perhaps surprising because this is approximately the age that VVD often becomes prominent

(*Julian, 1998*; *Julian, 2005*; *Randall & Mills, 1981*), although the timing of onset is known to vary (*Randall & Mills, 1981*).

Rotated Tibia (RT) occurred in birds at all developmental stages at a higher than expected rate (i.e., value) compared to previous work (0.2%, *Bradshaw, Kirkden & Broom, 2002*). However, Bradshaw et al., reported a reduced proportion of RT in older broilers, which perhaps indicates that affected birds were culled because the condition becomes clearly obvious around 21 days of age (*Riddell & Springer, 1985*). The exact aetiology of RT is unknown; however TD and VVD may exacerbate the incidence of RT (*Bradshaw, Kirkden & Broom, 2002*), thereby contributing to the relatively high proportion of birds with an outward torsion of the tibial shaft.

Overall, there is a clear need to monitor the leg health of flocks, not only to aid breeders to make adjustments to management practices or genetics when necessary, but also to quickly identify lame birds for euthanasia on welfare grounds. The high incidence of leg pathologies highlights the problem of maintaining high growth rates and breast muscle mass ($M_b$) at the expense of broiler anatomy and physiology. In addition, lameness represents a significant economic cost to the industry as birds with leg weakness are prematurely culled or have an increased incidence of mortality (*Mench, 2004*). Efforts to improve the health of growing broilers will have the twin benefit of improving both welfare standards and productivity.

## Muscle architecture

Architectural properties used to calculate the effective physiological cross-sectional area (PCSA) (*Gans & Bock, 1965*) of muscle take into account the effect of pennate fascicles on maximizing force per unit area. PCSA is thus greater in pennate muscles and is directly proportional to its force generation capacity (*Burkholder et al., 1994*; *Lieber & Friden, 2000*). In broilers, the PCSA of the major hip, knee and ankle extensors (essential for supporting body mass and maintaining an upright standing posture; *Gatesy, 1999*; *Reilly, 2000*; *Hutchinson, 2004*), scale with positive allometry — i.e., these muscles have a greater force-generating capacity (reflected in their relatively larger PCSAs) as the broiler develops (Table 4, Fig. 3). As a result, muscular force production capacity in broiler chickens should increase with age, likely as a direct consequence of weight vs. force scaling constraints imposed by resisting gravity and inertia (e.g., *Corr et al., 2003a*). However, these muscles still have smaller force-generating capabilities and shorter, presumably slower-contracting muscles than their wild counterpart, the Giant Junglefowl (*Paxton et al., 2010*). Broiler chickens appear to generally increase the PCSA of their pelvic limb muscles by increasing muscle mass, rather than by increasing fascicle length, which scaled isometrically (Table 4, Fig. 3). This increase in mass is likely due to increased hypertrophy (increase in muscle fibre size), which is well known to occur in broiler skeletal muscle and is the assumed dominant model for postnatal growth (*Aberle & Stewart, 1983*; *Soike & Bergmann, 1998*; *Remignon et al., 1994*; *Goldspink & Yang, 1999*). In addition, muscular force production is invariant to muscle fascicle length, but longer fascicles exact a metabolic cost because a larger volume of muscle is activated for each Newton of force (*Kram & Taylor, 1990*;

*Roberts, Chen & Taylor, 1998*). Thus, the isometric scaling of fascicle length we observed in this study avoids such added costs.

Interestingly, the PCSAs of the M. iliotibialis cranialis (IC) and M. tibialis cranialis caput femorale (TCF) scale more in accordance with isometry. The relative force-generating capacity of these muscles therefore remains unchanged throughout the growth of the broiler. The TCF is a knee extensor and ankle flexor and is assisted by other muscles that also serve as knee extensors and ankle flexors (e.g., M. femorotibialis and M. extensor digitorum longus). Similarly, the IC is also a knee extensor, but additionally acts as a hip flexor, supported by the M. iliotrochantericus caudalis (ITC; known to be significantly larger in the broiler *Paxton et al., 2010*) to flex and medially rotate the femur. The additional support of these muscles may help to explain why the IC and TCF scale in unusually isometric ways. The IC and TCF may be redundant, especially when limb motion in these broilers is likely to be (1) knee-driven (e.g., *Gatesy, 1999*; *Reilly, 2000*; *Hutchinson & Allen, 2008*), requiring action of the major knee flexors (*M. iliofibularis, M. flexor cruris lateralis*), and (2) three-dimensional, demanding large supportive forces at the hip for the considerable mediolateral forces they experience when they walk (*Paxton et al., 2013*). However, biomechanical analyses of *in vivo* function are needed to test how much their function alters with growth in broilers.

## Centre of mass and inertial properties

At the youngest age (14 days old) studied here, chickens' trunk mass accounted for ∼75% of total body mass. At slaughter weight (around 42 days of age), total trunk mass had increased to ∼80% total body mass. The ∼5% increase in trunk mass is largely attributable to pectoral muscle growth, which occurred at a relatively faster rate than body mass (see Part I, *Tickle et al., 2014*).

Interestingly, relative hind limb segment mass (summed segment masses; muscle and bone mass combined) did not decrease during growth, accounting for ∼15% of total body mass at slaughter age and thus representing a total 5% body mass increase across ontogeny. The proportion of bone mass contributing to total segment mass is likely small because the muscle to bone ratio is known to be high in commercial broilers (*Ganabadi et al., 2009*). The increase in leg mass was instead incurred by increases in drumstick and thigh muscle mass. Drumstick segment mass increased across ontogeny, becoming relatively larger at each age category, whereas thigh segment mass only had substantial changes during the last two weeks of growth (from 28 to 42 days old). Thigh muscle mass increased by ∼3% of total body mass during this period. Changes in thigh and drumstick segment mass are expected, as these segments yield the most meat and are the most consumed portions (alongside breast meat) on the market (*Broadbent, Wilson & Fisher, 1981*). However, the increase in hind limb segment mass is striking and comparable to a progenitor population (total limb muscle mass ∼16%) at the same approximate physiological mass and indeed larger than its wild counterpart at the same age by ∼4% body mass (see *Paxton et al., 2010*). Previous studies typically show an ontogenetic reduction in the investment of metabolic resources towards pelvic limb muscle growth (e.g., *Berri et al., 2007*;

*Schmidt et al., 2009*) and the main drivers of selection in broiler chickens are still a greater yield of breast muscle mass and a faster post-hatch growth rate (*Arthur & Alburs, 2003*). Thus, changes in leg muscle mass may not reflect a direct difference in selection pressures. However, a relative increase in hind limb muscle mass may reflect a functional demand for larger hip and knee extensors to support their increasing body mass. *Corr et al. (2003a)* studied two strains of birds (relaxed and selected) raised on two different feeding regimes and suggested that the large pectoral muscle mass of the broiler has displaced their CoM cranially. Similarly, *Abourachid (1993)* suggested that increased stresses on the pelvic limbs of heavier broad-breasted turkeys were induced by a more cranially positioned CoM. We found that broilers show a change in whole body CoM position consistent with these previous findings, shifting from a caudodorsal to a craniodorsal location between 28 and 42 days of age, which would increase demand for muscular force production to balance it. Interestingly, the centre of mass of the Giant Junglefowl (a representative progenitor population) has been shown to move caudodorsally, not craniodorsally, across ontogeny. This cranial shift in broilers may therefore be the direct result of increased pectoral muscle mass growth between four and six weeks of age.

In all cases, the craniocaudal CoM location in broilers was far more cranial than previous estimates. *Allen, Paxton & Hutchinson (2009)* estimated CoM cranial position to be ~38% of femur length, compared to the 70–90% femur length estimated in this study. On the other hand, dorsoventral estimates broadly corresponded to literature values (*Allen, Paxton & Hutchinson, 2009*). The difference in craniocaudal CoM position likely relates directly to pectoral muscle growth. The pectoral muscle mass of the broilers used in this study yielded an additional ~4% of total body mass in comparison to the broilers used by *Allen, Paxton & Hutchinson (2009)*. One limitation of our study is that whole body CoM was normalized by femur length, which did scale with slight negative allometry (see discussion below). Using femur length may introduce a slight bias to our results, but normalization by other factors including total limb length have been shown to yield the same result (*Allen, Paxton & Hutchinson, 2009*).

There are also substantial changes in the distal segment of the pelvic limb between 28 and 42 days. Foot mass increases by ~30% and foot CoM moves distally. Large feet may serve to improve the apparent stability reported in the broiler (e.g., *Corr et al., 2003b*) during the stance phase of gait. However, larger feet will influence the broiler's ability to accelerate and decelerate the limb during swing, which in turn can affect the metabolic cost of locomotion (*Kilbourne, 2013*; *Kilbourne & Hoffman, 2013*). Both an increase in mass and a more distal shift in the limb's mass distribution will effectively increase a limb's moment of inertia, (i.e., resistance to angular acceleration, *Steudel, 1990*; *Wickler et al., 2004*; *Kilbourne & Hoffman, 2013*). Broiler pelvic limbs would therefore require more metabolic energy to accelerate and decelerate them as the birds grow. However, the influence of mass is much smaller than the influence of the distribution of the mass (radii of gyration reported here) on the moment of inertia. Doubling mass would essentially double the moment of inertia, whereas doubling the radii of gyration would increase the moment of inertia four-fold (Eq. (3)). The radii of gyration in the parasagittal plane and

about the axes of long-axis rotation increased in the foot segments of the ∼42 day old broiler. Thus, increased radii of gyration in the foot segment of the broiler contribute significantly to the limb's moment of inertia.

We found that broilers' whole pelvic limb morphology changed across ontogeny, with the main changes in the thigh and foot segments. These segments had increased muscle mass and a more distal mass distribution (rather than having relatively longer limbs; see bone scaling discussion), resulting in relatively larger moments of inertia. The changes in limb morphology are likely to assist in supporting the increased supportive forces required by a more cranially positioned CoM and to help improve stability during locomotion. However, these changes also likely exact a relatively higher metabolic cost to locomotion.

### Bone scaling

We have shown that femur length scaled with slight negative allometry, whereas the lengths of the tibiotarsus and tarsometatarsus scale with isometry across broiler ontogeny (Fig. 4). Regardless, total relative limb length remained unchanged from four to six weeks in growing broilers may be an adaptation related to their apparent instability (*Paxton et al., 2013*). Maintaining shorter limbs may act to moderate the lateral motion of the CoM and aid balance (*Bauby & Kuo, 2000*). However, short limbs also likely lead to an increased energetic cost (*Steudel-Numbers & Tilken, 2004*).

Here we have considered how pelvic limb morphology changes during broiler ontogeny, in coordination with other changes such as pectoral muscle mass. Together, these changes have influenced broiler morphology across ontogeny, which may have influenced locomotor ability as well. The relative force-generating capacity of the hind limb muscles is greater in older broilers, and is primarily achieved through increasing muscle mass but maintaining a constant fascicle length. Increases in thigh segment mass and a relative increase in the moment of inertia of the distal limb (due mainly to increased foot size) may reflect adaptations to cope with the apparent instability and a more cranially positioned CoM as broilers grow. Although the architectural changes we have observed have obvious advantages for maintaining an upright posture and forward progression of broiler chickens, these morphological changes likely have a negative impact on locomotion, exacting relatively higher metabolic costs during growth, which may have knock-on consequences for activity levels and even overall health.

## ACKNOWLEDGEMENTS

We would like to thank Cobb-Vantress, Inc., especially Kate Barger and Antony Taylor for providing the chickens used in this study.

### Funding

This work was supported by a BBSRC grant (BB/I021116/1) to JRH and JRC. The funders had no role in study design, data collection and analysis, decision to publish, or preparation of the manuscript.

## Grant Disclosures

The following grant information was disclosed by the authors:
BBSRC: BB/I021116/1.

## Competing Interests

John Hutchinson is an Academic Editor for PeerJ. The authors declare there are no competing interests.

## Author Contributions

- Heather Paxton and Peter G. Tickle conceived and designed the experiments, performed the experiments, analyzed the data, contributed reagents/materials/analysis tools, wrote the paper, prepared figures and/or tables, reviewed drafts of the paper.
- Jeffery W. Rankin conceived and designed the experiments, performed the experiments, contributed reagents/materials/analysis tools, reviewed drafts of the paper.
- Jonathan R. Codd and John R. Hutchinson conceived and designed the experiments, contributed reagents/materials/analysis tools, reviewed drafts of the paper.

## Animal Ethics

The following information was supplied relating to ethical approvals (i.e., approving body and any reference numbers):

Full ethical approval for this experiment was granted by the RVC Ethics Committee (approval URN No. 2008-0001) under a Home Office License.

## Supplemental Information

Supplemental information for this article can be found online at http://dx.doi.org/10.7717/peerj.473.

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
