# Peer review of "Anatomical and biomechanical traits of broiler chickens across ontogeny. Part II. Body segment inertial properties and muscle architecture of the pelvic limb"

_PeerJ, doi:10.7717/peerj.473_

## Round 0.1 · original submission · Minor Revisions

· Academic Editor

Minor Revisions

In addition to the comments listed in the reviewers' reports, one reviewer indicated that "the authors do not provide data from less derived chickens which would allow an understanding of how much broilers have changed and in ways they have changed." So, please address this point (in addition to those listed in the reviewer's reports)in the revision. A list of point-to-point responses to the specific comments should also be provided.

Reviewer 1 ·

Basic reporting

This MS focusing on the pelvic limb reports on part 2 of a larger study on broiler chickens. Despite some references being made to part 1 of the study, the current MS is essentially self-contained. Nevertheless it is advantageous to have part 1 of the study (focusing on the respiratory system) available as a PeerJ Preprint. Overall, the authors have done a great job writing this MS. The introduction provides sufficient background to understand the research question, specific aims, and methodological approach. Results are presented clearly and all tables and figures are relevant. All conclusions are well supported by the data. I have noticed only some minor points that I would like to direct the authors’ attention to (see below).

Experimental design

The relevance of the research question addressed in this MS is obvious (welfare concerns of broiler chickens). All methods are clearly described and should enable other researchers to reproduce this study. From my experience, the number of specimens analyzed (5 per age class) is enough to capture most of the variability of anatomical parameters like muscle mass and average fascicle pennation angle and length. However, the authors also examined pathologies of the pelvic limbs and for this question the sample size might be too small to derive robust results. Consequently, the authors do not overstate their results on limb pathology. It is stated that ethical approval was granted, but no license number is provided. At least 40 chicken were sacrificed for this study. I would like the authors to consider detailing how chicken were killed. Were those the same cadavers that also were used in the first part of the study (Tickle et al. MS)?

Validity of the findings

The data collected in this study is robust and conclusions are well supported by this data.

Additional comments

Given the considerable space devoted to a discussion of the pathology of the limbs in broiler chicken in the MS, I was surprised to not see this aspect represented in the title or abstract. I would like the authors to consider to partly re-word the abstract: I suggest to not mention the consequences for ventilation, because this aspect is not discussed in the MS. Instead, findings concerning the pathology could be included. In the present form the final 2 sentences of the abstract read like a summary for both parts of the study, which from my understanding is not desired and every article should be self-contained.

Abstract, line 4: Please insert a space between ontogeny and influence.

Introduction, lines 94-104: Please note that more studies than are mentioned in the MS reported center of mass position and mass properties of avian limb and body segments to derive the body’s center of mass position (lapwing: Nyakatura et al., 2012; quail: Andrada et al., 2013).

M&M, line 114: Why have hatchlings not been included in the dataset for the calculation of center of mass across ontogeny?

M&M, lines 144-146: Where is the data for the fiber pennation angle shown? It is stated that the data for fascicle length and pennation angle was calculated as the mean of five measurements made across the muscle. Please consider including information on how variable this data is (how representative is the mean?).

M&M, lines 206-208: Please consider omitting the sentence starting with “However, …”.

Discussion, line 302: Given that different definitions of infection were used in previous studies and that the severity of an infection was not quantified in the current study, please consider clarifying the sentence starting with “BCO was common…”.

Discussion, lines 345, 346: The way “et al.” is written should be consistent throughout the MS.

Discussion, line 358: The statement “birds with leg weakness are prematurely culled or have an increased incidence of mortality” might need a reference.

Discussion, line 443: The paper of Kilbourne (2013 Biol J Linn Soc) might be relevant here as well.

Table 2: Please indicate whether the data for BCO is for presence (I assume) or absence of an infection.

Table 5: Please provide more information. What exactly does the superscript indicate? Relative to what was CoM position measured? Was CoM position assumed to be within the sagittal plane?

Reviewer 2 ·

Basic reporting

The manuscript is well written and very clear. The data are valuable and clearly presented.

Experimental design

This is a carefully conducted study that describes ontogenetic changes in the musculoskeletal system of the hind limb and the position of center of mass of broiler chickens. The analyses appear to be fully appropriate.
My only concern is that the manuscript provides very little comparative information that would allow a broader understanding of what the observed ontogenetic patterns represent. For example, it would be helpful to have similar anatomical measurements during ontogeny for junglefowl or outbred chickens. Such data are clearly beyond the scope of this study, but without this comparative information many of the more interesting questions that these data might address cannot be answered.

Validity of the findings

The findings represent an important contribution.

---

## Round 0.2 · accepted · Accept

· Academic Editor

Accept

Thanks for making the necessary revisions.